# Inner-Outer Aware Reconstruction Model for Monocular 3D Scene Reconstruction

**Yu-Kun Qiu**[1]      **Guo-Hao Xu**[1]      **Wei-Shi Zheng**[1,2] *

[1] School of Computer Science and Engineering, Sun Yat-sen University, China
[2] Key Laboratory of Machine Intelligence and Advanced Computing, Ministry of Education, China
`{qiuyk,xugh23}@mail2.sysu.edu.cn`
`wszheng@ieee.org`

## Abstract

Monocular 3D scene reconstruction aims to reconstruct the 3D structure of scenes based on posed images. Recent volumetric-based methods directly predict the truncated signed distance function (TSDF) volume and have achieved promising results. The memory cost of volumetric-based methods will grow cubically as the volume size increases, so a coarse-to-fine strategy is necessary for saving memory. Specifically, the coarse-to-fine strategy distinguishes surface voxels from non-surface voxels, and only potential surface voxels are considered in the succeeding procedure. However, the non-surface voxels have various features, and in particular, the voxels on the inner side of the surface are quite different from those on the outer side since there exists an intrinsic gap between them. Therefore, grouping inner-surface and outer-surface voxels into the same class will force the classifier to spend its capacity to bridge the gap. By contrast, it is relatively easy for the classifier to distinguish inner-surface and outer-surface voxels due to the intrinsic gap. Inspired by this, we propose the inner-outer aware reconstruction (IOAR) model. IOAR explores a new coarse-to-fine strategy to classify outer-surface, inner-surface and surface voxels. In addition, IOAR separates occupancy branches from TSDF branches to avoid mutual interference between them. Since our model can better classify the surface, outer-surface and inner-surface voxels, it can predict more precise meshes than existing methods. Experiment results on ScanNet, ICL-NUIM and TUM-RGBD datasets demonstrate the effectiveness and generalization of our model. The code is available at `https://github.com/YorkQiu/InnerOuterAwareReconstruction`.

## 1   Introduction

Monocular 3D Reconstruction is a fundamental task in computer vision and is the basis for various applications like robotics, augmented/virtual reality and autonomous navigation. It aims at reconstructing the 3D structure of scenes or objects based on a sequence of monocular RGB images. Although camera poses can be estimated accurately with state-of-the-art SLAM [1, 2] or SfM systems [3], this task is still very challenging since the complexity of the environment. For example, indoor scenes can contain various objects with different textures under diverse illumination.

Researchers have studied this challenging problem for many years, and conventional methods solve this problem in a multi-view depth manner [4, 5, 6, 7]. They predict the depth map for each image and then fuse these depth maps to generate 3D meshes. Recently, some researchers [8, 9, 10, 11, 12] attempted to regress the 3D surface of the entire scene directly and have achieved promising results. They first extract image features from different views and then back-project them to form volume

---

*Corresponding author.

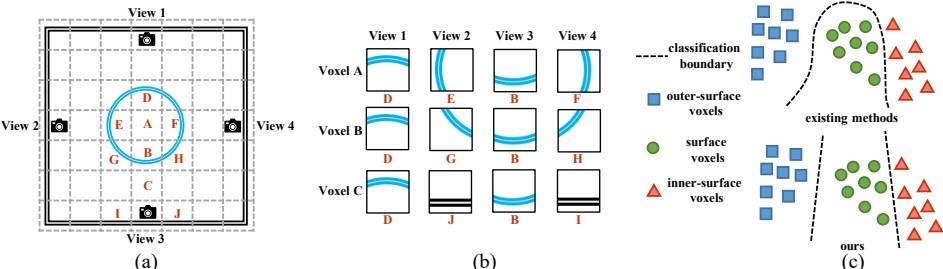

Figure 1: (a) Illustration of a simple scene with voxels (denoted by capital letters, *e.g.*, A, B and C). (b) The observations of voxel A, B and C in four different views. (c) The classification boundaries find by existing methods and our model.

features. Finally, they utilize a 3D backbone to predict a TSDF volume which can be converted to 3D mesh using the marching cubes algorithm [13]. As the volume size increases, the consumed memory will grow cubically. Therefore, existing methods apply a coarse-to-fine strategy to reduce memory costs. Specifically, they train a classifier for each level to classify whether a voxel is a surface voxel or not, and only the potential surface voxels will be selected and passed to the next level. However, the voxels on the inner side of the surface (inner-surface voxels) are quite different from the voxels on the outer side of the surface (outer-surface voxels). Due to the occlusion of the surface, the observations of inner-surface voxels are usually similar to the surface voxels. On the other hand, the outer-surface voxels are not occupied, and thus their observations from different views are surfaces of different instances. Hence, there is an intrinsic gap between the inner-surface voxels and outer-surface voxels. We give an example in Figure. 1. Figure. 1 (a) is a simple scene with only a circular object in the center of a room. And Figure. 1 (b) shows the observation of voxels A, B and C in four different views. As we can see, observations of inner-surface voxel A and outer-surface voxel C in views 2 and 4 are quite different (i.e., object surface vs. wall surface). In fact, the inner-surface voxel A is more similar to the surface voxel B since their observations under all views are different parts of the object's surface. Due to the large gap between inner-surface voxels and outer-surface voxels, it is improper to group all of them into one class, which requires the model to spend its capacity to bridge the gap. What is worse, this can lead to overfitting. As shown in Figure. 1 (c), previous methods have to learn complicated classification boundaries to classify the surface voxels and non-surface voxels.

By contrast, as there exists an intrinsic gap, classifying the inner-surface voxels and outer-surface voxels into two classes is relatively easy, meanwhile avoiding the waste of model capacity and alleviating the risk of overfitting. Inspired by this, we propose the inner-outer aware reconstruction (IOAR) model. Different from existing methods, IOAR explores a new coarse-to-fine strategy in which a classifier can classify whether a voxel is a surface, inner-surface or outer-surface voxel. In addition, we design separate TSDF and occupancy branches to avoid the mutual interference between TSDF prediction and occupancy prediction. Finally, we apply an inner-outer aware loss to supervise the inner-outer aware occupancy branch to learn to classify the inner-surface, outer-surface and surface voxels. Since it is easier to find classification boundaries between surface voxels and inner-surface voxels and between surface voxels and outer-surface voxels (Figure 1 (c)), IOAR can make more accurate predictions than previous methods. Experiments show that IOAR improves the F-score from 64.1 to 66.3 and the Precision from 70.8 to 74.8 on ScanNet. Further experiments on ICL-NUIM and TUM-RGBD datasets show that IOAR has good generalization performance.

## 2  Related Works

**Depth-based 3D reconstruction methods** reconstruct the 3D structure by two steps. First, estimate the depth of each selected video frame. Second, fuse these depths to reconstruct the 3D structure. Conventional methods [14, 15, 16, 17] typically use patch-matching based photometric consistency to estimate the depth. With the development of deep learning, many researchers paid attention to estimating accurate and dense depth maps using deep neural networks [6, 18, 5, 7, 4, 19, 20, 21]. MVSNet [6] extracts image features from multiple views and then builds a 3D cost volume based on the variance between reference view and source views. Then MVSNet predicts the depth map of the reference view based on the cost volume. MVDepthNet [5] builds the cost volume based on

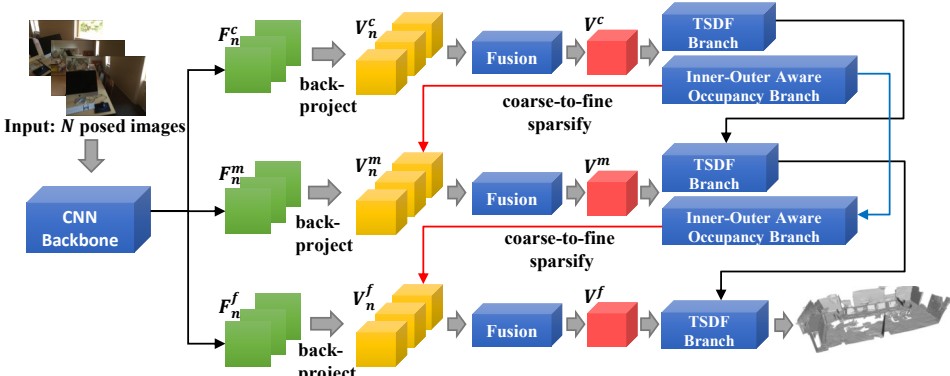

Figure 2: The overall pipeline of our model. Given a series of posed images, we first extract the coarse/medium/fine-level image features $F_n^c, F_n^m, F_n^f$ using a CNN backbone. Then, image features are back-projected to form feature volumes $V_n^c, V_n^m, V_n^f$. A fusion module is utilized to fuse feature volumes from different views. The fused features $V^c, V^m, V^f$ are then input into the TSDF or occupancy branch to predict TSDF or occupancy volume. The TSDF or occupancy features and logits are utilized to predict TSDF or occupancy volume in the next level.

pixel intensity and applies an encoder-decoder architecture to predict depth in multiple resolutions simultaneously. GP-MVS [7] utilizes the Gaussian process to leverage information from previous latent-space encodings to predict precise depth maps. Instead of directly predicting depth maps, Neural RGB→D [22] estimates a Depth Probability Volume (DPV) for each input frame. They design a deep network based on Bayesian filtering theory, which accumulates the DPVs across frames to predict accurate depth maps. DeepV2D [23] proposes a differentiable flow-to-depth layer and estimates the depth maps based on the optical flow. DeepVideoMVS [4] is an LSTM based model, which explores the scene geometry information from previous steps. SimpleRecon [19] injects readily available metadata into the cost volume and achieves good performance without the help of 3D convolutions. After estimating the depth, methods like Poisson surface reconstruction [24, 25], Delaunay Triangulation [26] and TSDF Fusion [27, 28, 29, 30] can be used to generate 3D mesh based on inconsistent single-view depths.

**Volumetric-based 3D reconstruction methods** directly regress a volumetric data structure to solve the problem in an end-to-end manner. SurfaceNet [31] builds colored voxel cubes (CVC) based on images and explores 3D convolutional networks to predict a surface occupancy volume. Atlas [8] back-projects the selected 2D image features into a 3D feature volume. Taking this volume as input, a 3D U-Net predicts a TSDF volume in a coarse-to-fine manner. Each voxel in the TSDF volume describes the truncated signed distance to the nearest surface and the TSDF volume can be converted to 3D mesh using the marching cubes algorithm [13]. NeuralRecon [9] achieves real-time 3D reconstruction by reconstructing the scene in each local window and then merges the result using a GRU [32] module. Instead of averaging the contribution of each image feature, TransformerFusion [10] pays attention to the most relevant images using a transformer mechanism. VoRTX [11] also applies a transformer mechanism to aggregate image features. In addition, VoRTX weights the contribution of image features for each voxel based on the projective occupancy. 3D-Former [12] designs an 3D Transformer to replace the 3D CNN in previous works and clearly improves the performance. Theoretically, 3D-Former can be utilized as the 3D backbone of any volumetric-based 3D reconstruction method, and our method can significantly improve performance with its help. However, to have a fair comparison with existing works, we still use the 3D CNN as the 3D backbone.

**Neural implicit representation of 3D structure.** Some researchers [33, 34, 35, 36] aim to find a neural implicit representation of the 3D structure. Mescheder *et al.* [33] present OccupancyNetwork which learns a mapping from 3D coordinates and observations to the probability of occupancy. Park *et al.* [34] propose DeepSDF which predicts the SDF value of each point. NSVF [35] progressively sparsifies the space and finds mapping in the sparse space, thus highly reducing the computational cost. NeRFusion [36] combines the advantages of NeRF and TSDF-based fusion methods to achieve efficient large-scale reconstruction.

In this work, we propose the inner-outer aware reconstruction (IOAR) model which aims to predict meshes more precisely by finding better classification boundaries between surface voxels and inner-surface voxels and between surface voxels and outer-surface voxels. IOAR explores a new coarse-to-fine strategy in which the classifier can classify the inner-surface, outer-surface, and surface voxels. In addition, IOAR avoids the mutual interference between TSDF and occupancy prediction by using separate branches to predict TSDF and occupancy volume. Under the supervision of inner-outer aware loss, the occupancy branch can predict the potential surface more accurately. Extensive experiments on several datasets demonstrate the effectiveness and generalization of our IOAR.

# 3 Inner-Outer Aware Reconstruction

Given a set of $N$ posed images $\{I_n\}_{n=1}^N$ of a scene with corresponding camera intrinsics $\{K_n\}_{n=1}^N$ and camera extrinsics $\{P_n\}_{n=1}^N$, our model aims to predict a TSDF volume $V_{TSDF} \in [-1, 1]^{N_X^f \times N_Y^f \times N_Z^f}$. This TSDF volume will further be converted to 3D mesh using the marching cubes algorithm [13]. We illustrate the pipeline of our model in Figure. 2. The posed images are input into a CNN backbone to extract image features at different scales. The feature of the image in view $n$ at coarse/medium/fine level are indicated as $F_n^c, F_n^m, F_n^f$. Then, these features are back-projected to form feature volumes $V_n^c, V_n^m, V_n^f$ and a fusion module fuses volumes from $N$ views to generate fused feature $V^c, V^m, V^f$. Finally, the fused features are input into the TSDF or occupancy branch to predict the TSDF or occupancy volume. In the medium and fine level, the TSDF/occupancy features and logits from the previous level are utilized to predict the TSDF/occupancy volume at current level.

## 3.1 Feature Volume Construction

The posed images are input into a 2D CNN backbone $\mathcal{F}(\cdot)$ to extract 2D features at different scales. Using $\mathcal{F}(\cdot)$, we can get the image feature of $I_n$ at coarse/medium/fine level $F_n^c, F_n^m, F_n^f = \mathcal{F}(I_n)$. Here, $F_n^c \in \mathbb{R}^{C_{in}^c \times H^c \times W^c}, F_n^m \in \mathbb{R}^{C_{in}^m \times H^m \times W^m}, F_n^f \in \mathbb{R}^{C_{in}^f \times H^f \times W^f}$. For simplicity, we focus on a single resolution $r$ ($r \in \{c, m, f\}$). Then these image features are back-projected to form feature volumes. Specifically, using corresponding camera intrinsic $K_n$ and camera extrinsics $P_n$ we can form the feature volume $V_n^r \in \mathbb{R}^{C_{in}^r \times N_x^r \times N_y^r \times N_z^r}$:

$$V_n^r(:, i, j, k) = F_n^r(:, \hat{i}, \hat{j}), \begin{bmatrix} \hat{\hat{i}} \\ \hat{\hat{j}} \end{bmatrix} = \Pi K_n P_n \begin{bmatrix} i \\ j \\ k \\ 1 \end{bmatrix}. \tag{1}$$

Here, $\Pi$ is the perspective mapping, : is the slice operator and $N_x^r \times N_y^r \times N_z^r$ is the volume size in resolution $r$. Eq. (1) maps the voxel coordinates $(i, j, k)$ in the world space to the pixel coordinates $(\hat{i}, \hat{j})$ in the image space and assigns the image features to the corresponding voxel features. This means that all voxels along a ray are filled with the same features of the corresponding pixel.

Then we need to fuse the feature volumes among all views. A straightforward method is averaging all feature volumes (*i.e.*, $V^r = \frac{1}{N} \sum_{n=1}^N V_n^r$). However, different views provide different information when predicting TSDF or occupancy of voxels. Therefore, assigning larger weights to views that provide information that contributes to TSDF or occupancy prediction results in better performance. Following VoRTX [11], we use a transformer to fuse the feature volumes and assign weights based on projective occupancy. First, a transformer [37] $\mathcal{T}^r(\cdot)$ is used to augment each single-view feature volume with information from other relevant views:

$$\hat{v}_1^r, \hat{v}_2^r, \cdots \hat{v}_N^r = \mathcal{T}^r(v_1^r, v_2^r, \cdots v_N^r). \tag{2}$$

Here, $v_n^r$ is the feature of a voxel in volume $V_n^r$ (*i.e.*, $v_n^r \in V_n^r$). Then, we can predict the projective occupancy $\hat{O}_v^r \in \mathbb{R}^N$:

$$\hat{O}_v^r = \sigma(W_v^r), W_v^r = \phi^r([\hat{v}_1^r, \hat{v}_2^r, \cdots, \hat{v}_N^r]). \tag{3}$$

Here, $\phi^r(\cdot)$ is an MLP and $\sigma(\cdot)$ is a sigmoid function. The corresponding ground-truth $(O_v^r)_n$ is equal to 1 if the voxel $v$ at view $n$ with resolution $r$ is close enough to the surface. The detailed definition of $O_v^r$ can be found in the supplementary material. We concatenate a zero to $W_v^r$ to indicate voxel $v$ is not close to surface under all $N$ views and apply a softmax to compute a weight vector $\hat{W}_v^r \in \mathbb{R}^{N+1}$.

Then the weighted voxel feature of voxel $v$ is computed by $v^r = \hat{W}_v^r[\hat{v}_1^r, \hat{v}_2^r, \cdots, \hat{v}_N^r, \vec{0}]$. After weighting all voxels, we can get the fused feature volume $V^r \in \mathbb{R}^{C_{in}^r \times N_x^r \times N_y^r \times N_z^r}$.

## 3.2 TSDF and Inner-Outer Aware Occupancy Branch

Taking the fused feature volume $V^r$ as input, the TSDF and inner-outer aware occupancy branch predict the TSDF value and occupancy probability for each voxel. The TSDF/occupancy branch at each level consists of a 3D CNN backbone and a TSDF/occupancy head.

**The procedure at the coarse level.** We denote the 3D CNN backbone of the TSDF/occupancy branch as $\psi_{TSDF}^c(\cdot), \psi_{OCC}^c(\cdot)$. The fused feature volume $V^c$ is input into each 3D CNN backbone and generates the TSDF feature $F_{TSDF}^c \in \mathbb{R}^{C_{out}^c \times N_x^c \times N_y^c \times N_z^c}$ and occupancy feature $F_{OCC}^c \in \mathbb{R}^{C_{out}^c \times N_x^c \times N_y^c \times N_z^c}$ by:

$$F_{TSDF}^c = \psi_{TSDF}^c(V^c), F_{OCC}^c = \psi_{OCC}^c(V^c). \tag{4}$$

The TSDF head $\mathcal{H}_{TSDF}^c(\cdot)$ is a 3D convolutional layer with the output channel equal to 1. Different from previous works, since we aim to classify the surface, inner-surface and outer-surface voxels, the occupancy head $\mathcal{H}_{OCC}^c(\cdot)$ is a 3D convolutional layer with an output channel equal to 3. We can get the TSDF logits $l_{TSDF}^c = \in \mathbb{R}^{1 \times N_x^c \times N_y^c \times N_z^c}$ and occupancy logits $l_{OCC}^c \in \mathbb{R}^{3 \times N_x^c \times N_y^c \times N_z^c}$ at the coarse level by:

$$l_{TSDF}^c = \mathcal{H}_{TSDF}^c(F_{TSDF}^c), l_{OCC}^c = \mathcal{H}_{OCC}^c(F_{OCC}^c). \tag{5}$$

**Inner-outer aware coarse-to-fine strategy.** Before introducing the procedure in the medium and fine level, we first introduce the inner-outer aware coarse-to-fine strategy utilized by the inner-outer aware occupancy branch. Given the ground-truth TSDF $y_{TSDF}^r \in \mathbb{R}^{N_x^r \times N_y^r \times N_z^r}$, we can define surface voxels, inner-surface voxels and outer-surface voxels as follows. A voxel $v$ is a surface voxel if it is very close to the surface (*i.e.*, this voxel is in the truncated range of a surface ($-1 < (y_{TSDF}^r)_v < 1$)). A voxel $v$ is an inner-surface voxel if it is on the inner side of the surface (*i.e.*, this voxel is not in the truncated range of any surface and its truncated distance is 1 ($(y_{TSDF}^r)_v = 1$)). A voxel $v$ is an outer-surface voxel if it is on the outer side of the surface (*i.e.*, this voxel is not in the truncated range of any surface and its truncated distance is -1 ($(y_{TSDF}^r)_v = -1$)). Then based on the definition of surface voxels, inner-surface voxels and outer-surface voxels, the occupancy ground-truth $y_{OCC}^r \in \mathbb{R}^{N_x^r \times N_y^r \times N_z^r}$ can be defined as:

$$(y_{OCC}^r)_v = \begin{cases} 0 & \text{if } v \text{ is a inner-surface voxel } (i.e., (y_{TSDF}^r)_v = 1), \\ 1 & \text{if } v \text{ is a surface voxel } (i.e., -1 < (y_{TSDF}^r)_v < 1), \\ 2 & \text{if } v \text{ is an outer-surface voxel } (i.e., (y_{TSDF}^r)_v = -1). \end{cases} \tag{6}$$

Given the occupancy probability at the coarse level $p_{OCC}^c = \text{softmax}(l_{OCC}^c)$, we can get the occupancy state at coarse level $s_{OCC}^c = \arg\max(p_{OCC}^c) \in \{0, 1, 2\}^{N_x^c \times N_y^c \times N_z^c}$. Intuitively, we can get an occupancy mask $mask^c$ at coarse level where $mask_v^c = \mathbb{I}((s_{OCC}^c)_v = 1)$. Here, $\mathbb{I}(c)$ equals to 1 if condition $c$ is satisfied. With the help of the occupancy mask, we can sparsify the dense volume and reduce memory and computation costs. All sparse operations in our work are based on torchsparse [38]. Different from existing methods, our coarse-to-fine strategy makes it easier for the classifier to find better classification boundaries which leads to precise mesh prediction. For simplicity, we still use dense volumes in the following description.

**The procedure at the medium/fine level.** Different from the coarse level, the features from the previous level are utilized for better prediction in the medium and fine levels. To utilize the features in the previous level, we expand TSDF/occupancy features and logits in the previous level to have the same size as the current level. The expanded TSDF features $\hat{F}_{TSDF}^c \in \mathbb{R}^{C_{out}^c \times N_x^m \times N_y^m \times N_z^m}$ and logits $\hat{l}_{TSDF}^c \in \mathbb{R}^{1 \times N_x^m \times N_y^m \times N_z^m}$ are concatenated with fused feature volume $V^m$ at the current level (medium level) to form medium level aggregated TSDF feature volume $V_{TSDF}^m \in \mathbb{R}^{(C_{out}^c + C_{in}^m + 1) \times N_x^m \times N_y^m \times N_z^m}$ which is used for extracting medium level TSDF features $F_{TSDF}^m \in \mathbb{R}^{C_{out}^m \times N_x^m \times N_y^m \times N_z^m}$:

$$F_{TSDF}^m = \psi_{TSDF}^m(V_{TSDF}^m), V_{TSDF}^m = \text{cat}(\hat{F}_{TSDF}^c, V^m, \hat{l}_{TSDF}^c). \tag{7}$$

Here, $N_x^m = 2N_x^c, N_y^m = 2N_y^c, N_z^m = 2N_z^c$. The procedure of generating TSDF logits $l_{TSDF}^m$ is the same as coarse level using TSDF head $\mathcal{H}_{TSDF}^m(\cdot)$. Similarly, we can extract the medium-level

Table 1: Evaluation of the 3D meshes on ScanNet. The upper part follows the evaluation protocol in VoRTX [11] and the lower part follows the evaluation protocol in TransformerFusion [10].

| Methods | Acc ↓ | Comp ↓ | Prec ↑ | Recall ↑ | F-score ↑ |
|---|---|---|---|---|---|
| DeepVideoMVS [4] | 0.079 | 0.133 | 0.521 | 0.454 | 0.474 |
| NeuralRecon [9] | 0.054 | 0.128 | 0.684 | 0.479 | 0.562 |
| Atlas [8] | 0.068 | 0.098 | 0.640 | 0.539 | 0.583 |
| VoRTX [11] | 0.054 | 0.090 | 0.708 | 0.588 | 0.641 |
| Ours | **0.043** | **0.090** | **0.748** | **0.597** | **0.663** |
| COLMAP [14] | 0.102 | 0.119 | 0.509 | 0.474 | 0.489 |
| MVDepthNet [5] | 0.129 | 0.083 | 0.443 | 0.487 | 0.460 |
| GP-MVS [7] | 0.129 | 0.080 | 0.453 | 0.510 | 0.477 |
| DPSNet [18] | 0.119 | 0.076 | 0.474 | 0.519 | 0.492 |
| ESTDepth [39] | 0.127 | 0.075 | 0.456 | 0.542 | 0.491 |
| DeepVideoMVS [4] | 0.107 | 0.069 | 0.541 | 0.592 | 0.563 |
| NeuralRecon [9] | 0.051 | 0.091 | 0.630 | 0.612 | 0.619 |
| TransformerFusion [10] | 0.055 | 0.083 | 0.728 | 0.600 | 0.655 |
| SimpleRecon [19] | 0.055 | **0.061** | 0.686 | 0.658 | 0.671 |
| Atlas [8] | 0.072 | 0.076 | 0.675 | 0.605 | 0.636 |
| VoRTX [11] | 0.043 | 0.072 | 0.767 | **0.651** | 0.703 |
| Ours | **0.037** | 0.072 | **0.791** | 0.650 | **0.712** |

occupancy feature $F_{OCC}^m \in \mathbb{R}^{C_{out}^m \times N_x^m \times N_y^m \times N_z^m}$ and the rest of the procedure is the same as the coarse level. A detailed description of the expansion operation is introduced in the supplementary material.

At the fine level, all voxels are treated as potential surface voxels. Therefore, we do not set a fine-level occupancy branch. Except for this, the procedure of the fine level is the same as that of the medium level.

### 3.3 Training Loss

The loss function can be divided into three parts, the projective occupancy loss, the inner-outer aware occupancy loss, and the TSDF loss. Following previous methods, the projective occupancy loss $\mathcal{L}_P^r$ is a binary cross-entropy loss [11] and the TSDF loss $\mathcal{L}_{TSDF}^r$ is a $l_1$ distance loss [8]. The inner-outer aware occupancy loss can be computed by:

$$\mathcal{L}_{OCC}^r = -\frac{1}{N_x^r \times N_y^r \times N_z^r} \sum_{v \in p_{OCC}^r} \log(p_{OCC_v}^r) y_{OCC_v}^r. \tag{8}$$

Here, $p_{OCC_v}^r$ is the occupancy probability of voxel $v$ at resolution $r$, $y_{OCC_v}^r$ is the ground-truth occupancy label of voxel $v$ at resolution $r$. By summing all losses, the total loss is:

$$\mathcal{L} = \mathcal{L}_P^c + \mathcal{L}_P^m + \mathcal{L}_P^f + \mathcal{L}_{OCC}^c + \mathcal{L}_{OCC}^m + \mathcal{L}_{TSDF}^f. \tag{9}$$

## 4 Experiments

**Datasets.** Following previous works [8, 9, 11], we trained models on the ScanNet [40] dataset. ScanNet is a large-scale indoor scene dataset, which consists of 1613 RGB-D videos of 806 scenes. We follow the official train/eval/test split, where 1201 videos are used for training, 312 videos are used for evaluating and 100 videos are used for testing. Previous works evaluate the generalization performance of the models trained on ScanNet on TUM-RGBD [41] and ICL-NUIM [42] datasets. TUM-RGBD and ICL-NUIM datasets are also composed of RGB-D videos. Following previous works [11], 13 scenes of TUM-RGBD and 8 scenes of ICL-NUIM are used.

**Evaluation metrics and protocol.** Following conventional settings [9, 8, 11], we evaluate the performance using both 3D reconstruction metrics (*e.g.*, Precision, Recall and F-Score) and 2D depth metrics (*e.g.*, RMSE, Comp and Abs Rel). The detailed definition can be found in the supplementary material. Conventionally, Precision, Recall and F-Score are the three most important metrics. There

Table 2: Evaluation of the 2D depth map on ScanNet.

| Methods | Abs Rel ↓ | Abs Diff ↓ | Sq Rel ↓ | RMSE ↓ | $delta - 1.25$ ↑ |
|---|---|---|---|---|---|
| COLMAP [14] | 0.137 | 0.264 | 0.138 | 0.502 | 0.834 |
| MVDepthNet [5] | 0.098 | 0.191 | 0.061 | 0.293 | 0.896 |
| GP-MVS [7] | 0.130 | 0.239 | 0.339 | 0.472 | 0.906 |
| DPSNet [18] | 0.087 | 0.158 | 0.035 | 0.232 | 0.925 |
| Atlas [8] | 0.065 | 0.124 | 0.043 | 0.251 | 0.936 |
| NeuralRecon [9] | 0.065 | 0.106 | 0.031 | 0.195 | 0.948 |
| VoRTX [11] | 0.061 | 0.096 | 0.038 | 0.205 | 0.943 |
| Ours | **0.052** | **0.082** | **0.031** | **0.182** | **0.958** |

Table 3: Evaluation of the generalization. All models are trained on ScanNet datasets and tested on ICL-NUIM and TUM-RGBD.

| Datasets | Methods | Acc ↓ | Comp ↓ | Prec ↑ | Recall ↑ | F-score ↑ |
|---|---|---|---|---|---|---|
| ICL-NUIM | Atlas [8] | 0.175 | 0.314 | 0.280 | 0.194 | 0.229 |
| | NeuralRecon [9] | 0.215 | 1.031 | 0.214 | 0.036 | 0.058 |
| | VoRTX [11] | 0.102 | 0.146 | 0.449 | 0.375 | 0.408 |
| | Ours | **0.064** | **0.112** | **0.607** | **0.437** | **0.507** |
| TUM-RGBD | Atlas [8] | 0.208 | 2.344 | 0.360 | 0.089 | 0.132 |
| | NeuralRecon [9] | 0.130 | 2.528 | 0.382 | 0.075 | 0.115 |
| | VoRTX [11] | 0.175 | 0.314 | 0.280 | 0.194 | 0.229 |
| | Ours | **0.072** | **0.140** | **0.564** | **0.248** | **0.343** |

are two different evaluation protocols. The original protocol directly compared the predicted 3D meshes with the ground-truth 3D meshes. Bozic *et al.*[10] notice that a more complete reconstruction can be penalized since some part of the ground-truth 3D meshes is incomplete. To avoid this problem, they applied an occupancy mask when evaluating models. In the following sections, we will conduct experiments to evaluate our framework under both protocols to demonstrate its effectiveness.

**Training Details.** We use the Adam optimizer [43] with $\beta_1 = 0.9, \beta_2 = 0.999$ and $\epsilon = 10^{-8}$. The learning rate is set to $\alpha = 10^{-3}$ and is linearly warmed up from $10^{-10}$ over 2000 steps. We trained our model for 500 epochs. We use the MnasNet-B1 [44] as the CNN backbone. We also apply the feature pyramid network [45] to extract features at different scales. The CNN backbone is fixed in the first 350 epochs and is finetuned with a learning rate $\alpha = 10^{-4}$ in the last 150 epochs. The batch size is set to 4 and drops to 2 in the finetuning stage. Following previous methods [9, 11], we randomly drop out voxels to reduce the memory cost. In the training stage, we select 20 views for each video and the selection strategy is following [9]. To have a fair comparison, the voxel size of the fine/medium/coarse level is set to 4cm/8cm/16cm and the TSDF truncation distance is set to triple the voxel size. Training our model takes about 90 hours on a single Nvidia RTX 3090 graphic card.

## 4.1 Evaluation

To evaluate the performance of our model, we first compare the performance of our model and existing state-of-the-art methods on the large-scale indoor scene dataset ScanNet. The results are reported in Table. 1. As shown in the table, our model achieves state-of-the-art performance on almost all metrics. For example, our model improves the F-score from 0.641 to 0.663 and the Precision from 0.708 to 0.748 when evaluated based on the first protocol [11]. Based on the Second evaluation protocol [10], our model can also increase the F-score from 0.703 to 0.712 and the Precision from 0.767 to 0.791. The design of our model aims to find better classification boundaries, which can lead to precise surface voxel prediction. The significant improvement in Precision demonstrates the effectiveness of our design and shows that our model can make much more precise predictions than existing methods.

Following previous works, we also evaluate our model with 2D depth map metrics. Since volumetric-based methods directly predict TSDF volume instead of depth maps, we render the predicted 3D meshes to the image planes to get the depth maps. The results are reported in Table. 2. The results

Table 4: Evaluation of each component of our model. Here, "Branch" indicates using separate branches to predict TSDF and occupancy. "IOA" indicates using the inner-outer aware coarse-to-fine strategy and loss.

| Branch | IOA | Acc ↓ | Comp ↓ | Prec ↑ | Recall ↑ | F-score ↑ |
|--------|-----|-------|--------|--------|----------|-----------|
| ✗ | ✗ | 0.054 | 0.090 | 0.708 | 0.588 | 0.641 |
| ✓ | ✗ | 0.054 | **0.086** | 0.712 | **0.607** | 0.654 |
| ✗ | ✓ | 0.044 | 0.092 | 0.743 | 0.583 | 0.652 |
| ✓ | ✓ | **0.043** | 0.090 | **0.748** | 0.597 | **0.663** |

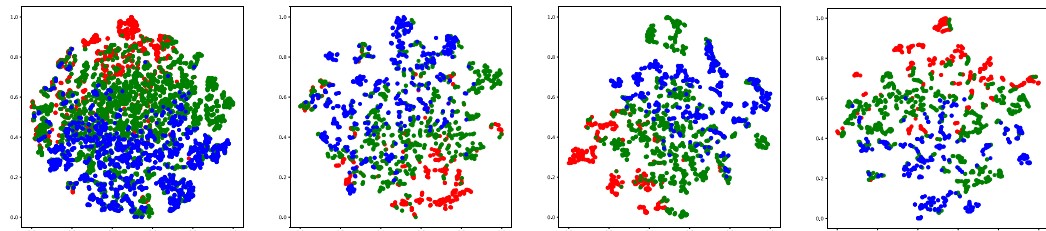

Figure 3: Visualizations of outer-surface, surface, and inner-surface voxels using t-SNE [46]. Here, the outer-surface, surface, and inner-surface voxels are colored by red, green and blue, respectively. Four subfigures are the results of four different scenes.

show that our model can decrease the relative error by 14.7% from 0.061 to 0.052 and get a more accurate depth map.

To evaluate the generalizability of our model, we also conduct experiments on two other datasets ICL-NUIM and TUM-RGBD. Models are trained on ScanNet and tested on ICL-NUIM and TUM-RGBD. The results are reported in Table. 3. On the ICL-NUIM dataset, the Precision is improved from 0.449 to 0.607 and the F-score is improved from 0.408 to 0.507. On the TUM-RGBD dataset, the Precision is improved from 0.280 to 0.564 and the F-score is improved from 0.229 to 0.343. Compared to existing methods, our model generalizes well on both datasets. A possible reason is that existing methods overfit the training set to find the complicated classification boundary between surface and non-surface voxels. Instead, our model finds the classification boundaries between inner-surface and outer-surface voxels, between surface and inner-surface voxels, and between surface and outer-surface voxels, which is much easier and thus alleviates overfitting.

## 4.2 Ablation Study

In addition to comparing our model with existing methods, we also conduct an ablation study to further explore the contribution of each part of our model. To this end, we design two variant models. The first one is using separate occupancy branches without the inner-outer aware coarse-to-fine strategy and loss. The second one uses the inner-outer aware coarse-to-fine strategy and loss but extracts occupancy and TSDF using a shared 3D CNN backbone. The results are reported in Table. 4. We separate the occupancy and TSDF branches since they may interfere with each other with a shared 3D CNN backbone. Compared to the basic model, using separate occupancy and TSDF branches can slightly improve all metrics. The TSDF prediction and occupancy prediction are both improved which demonstrates the effectiveness of this design. The inner-outer aware coarse-to-fine strategy and loss aim to better classify the surface voxels. Compared to the basic model, using inner-outer aware coarse-to-fine strategy and loss leads to significant improvement in Precision. This result shows that using the new strategy and loss can help the model make more precise predictions and better classify the surface voxels. Finally, the full IOAR model can further improve the performance, which shows that the two components can collaborate well.

## 4.3 Further Analysis

In this section, we further analyze our model with qualitative and quantitative results. First, we answer an important question: are inner-surface voxels really quite different from outer-surface voxels?

| Atlas | NeuralRecon | VoRTX | Ours | Ground-Truth |
|---|---|---|---|---|

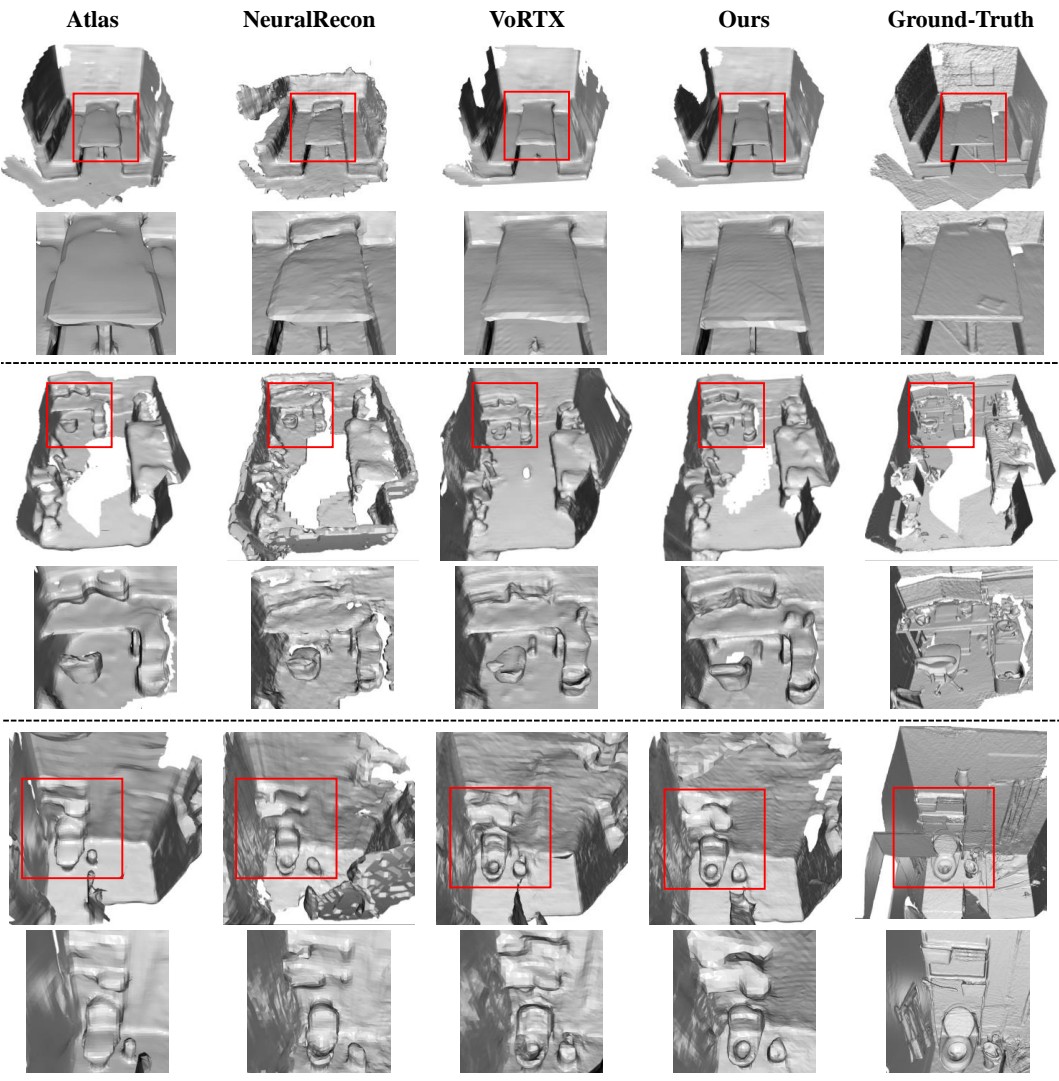

Figure 4: Visualization of 3D meshes reconstructed by our model and existing methods. From left to right are the meshes reconstructed by Atlas, NeuralRecon, VoRTX and our model. And the most right column contains the ground-truth meshes. We enlarge the detailed area in the red box to display it more clearly.

Second, we show some visualization results to compare the quality of 3D meshes reconstructed by our model and other methods.

**Are inner-surface voxels really quite different from outer-surface voxels?** This is an important question since grouping inner-surface and outer-surface voxels into the same class will force the model to waste its capability only when there is a gap between the inner-surface and outer-surface voxels (i.e., the inner-surface voxels are quite different from the outer-surface voxels). To answer this question, we conduct both qualitative and quantitative experiments. First, we visualize the features of surface, inner-surface, and outer-surface voxels in Figure. 3. As we can see, the features of inner-surface and outer-surface voxels are quite different and the features of surface voxels usually distribute between inner-surface and outer-surface voxel features. Therefore, it is difficult for a classifier to distinguish non-surface voxels from surface voxels, which leads to a complicated classification boundary. This supports the motivation of our model. Second, we measure the similarity between outer-surface voxels and inner-surface voxels (denoted as $s_{oi}$), the similarity between outer-surface voxels and surface voxels (denoted as $s_{os}$) and the similarity between inner-surface voxels and surface voxels (denoted as $s_{is}$). Since there are a large number of voxels in each volume, measuring

the similarity between all pairs of voxels is computationally impossible. Instead, we compute the feature prototypes of surface, non-surface, inner-surface and outer-surface voxels by averaging the features. Then, we measure the similarity between each class based on the cosine similarity between prototypes. We average the similarity over all test scenes and get $s_{oi} = 0.6956$, $s_{os} = 0.8933$ and $s_{is} = 0.8829$. This also demonstrates that inner-surface voxels are quite different from outer-surface voxels. We also analyzed the accuracy of discriminating inner-surface, outer-surface and surface voxels. The results show that only a few inner-surface voxels are misclassified as outer-surface voxels and only a few outer-surface voxels are misclassified as inner-surface voxels. This also supports our assumption that outer-surface voxels are quite different from inner-surface voxels since the model can easily distinguish them in most cases. For more details about this experiment, please refer to our supplementary material.

**Visualization.** To have an intuitive comparison with existing works, we visualize the 3D meshes reconstructed by our model and other methods in Figure. 4. As we can see, the reconstructed meshes of our model are more precise and complete than existing methods. Taking the first scene as an example, our model can precisely reconstruct a flat desktop while the desktop reconstructed by Atlas and NeuralRecon are uneven. And compared with VoRTX, we can reconstruct the table leg which is lost in their result.

## 5   Conclusion

In this work, we propose the inner-outer aware reconstruction (IOAR) model for monocular 3D scene reconstruction. Different from existing methods, IOAR can classify the inner-surface voxels and outer-surface voxels, which contributes to precise occupancy prediction and TSDF prediction. To achieve this goal, we design new inner-outer aware coarse-to-fine strategy and loss, which leads the model to learn to classify the inner-surface voxels and outer-surface voxels. In addition, to avoid the mutual interference between TSDF prediction and occupancy prediction, IOAR separates the occupancy branches from the TSDF branches. Thanks to these effective designs, IOAR achieves state-of-the-art performance on the large-scale indoor scene dataset ScanNet and generalizes well on unseen scenes.

## Acknowledgement

This work was supported partially by the NSFC(U21A20471,U1911401), Guangdong NSF Project (No. 2023B1515040025, 2020B1515120085).

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
