# OpenReview forum: "Inner-Outer Aware Reconstruction Model for Monocular 3D Scene Reconstruction"
_NeurIPS.cc/2023/Conference — NeurIPS 2023 poster_

### Official Review · Reviewer_Ga1p · 2023-07-03

**Soundness:** 3 good
**Presentation:** 2 fair
**Contribution:** 3 good
**Rating:** 6
**Confidence:** 4

**Summary:**

The authors propose a method for 3D reconstruction given images and the respective camera poses. Existing methods predict a TSDF volume and convert to 3D mesh using marching cubes. However, the authors proposed method as a coarse to fine strategy which uses a classifier to classify a voxel into surface, inner-surface or outer-surface voxel. Doing this they achieved better performance on 3D reconstruction

**Strengths:**

- The main strength of the paper is the proposed coarse-to-fine strategy which aids in the reduction of the memory costs of reconstruction and better performance on the reconstructed 3D meshes.
- The method has been evaluated on reconstruction datasets such as ScanNet, ICL-NUIM, and TUM-RGBD
- Code has been provided for reference. (I did not run it but have referred it when referring to the write up)
- The related work has referenced all the relevant works in the area.

**Weaknesses:**

- The impact of classifying the difference between inner surface voxels and outer surface voxel is not entirely clear. Especially since the claim is that the classification of vowels into different classes brings major gains. Section 4.3 provides write up on this but it is not entirely clear from the write up cosine similarity between the average of features of voxels is providing demonstrable differences.

Please address the typos :
1. Line 86 predicts*
2. Line 89 merges*
3. Line 177 get an*
3. Line 183 medium/fine*
4. Line 189 extracting*
5. Line 193 rest of the procedure*
6. Line 286 Since there are*
7. Line 304 What is more -> In addition,

There are further typos in the supplementary section which I have not listed here.

**Questions:**

- Perhaps section 4.3 could be written better?
- Could you provide details on the inference time etc.,?

**Limitations:**

The method is subject to limitations by occlusion and partial scans along with transparent surfaces as listed in Section B (supplementary)

---

> ### Author Rebuttal · Authors · 2023-08-09
>
> **Q1: The impact of classifying the difference between inner surface voxels and outer surface voxels is not entirely clear.**
>
> **Re:** We showed the impact of classifying the difference between inner surface voxels and outer surface voxels with qualitative experiment results in the ablation study (Section 4.2). The experiment results in Table 4 showed that IOAR (line 4 of Table 4) outperforms the IOAR variant that groups the inner surface voxels and outer surface voxels into the same class (line 2 of Table 4). Specifically, the precision improves from 0.712 to 0.748, and the F-score improves from 0.654 to 0.663 when classifying the difference between inner surface voxels and outer surface voxels. This experiment result demonstrates that classifying the difference between inner surface voxels and outer surface voxels can lead to clear performance improvement.
>
>  **Q2: The authors write section 4.3 to show the difference between inner surface voxels and outer surface voxels, but it is not entirely clear. Perhaps it could be written better?**
>
> **Re:** Thank you for your comment! We summarize the content in Section 4.3 in the following. If you are confused about any part, please question us for more details. In the ablation study (Section 4.2), we experimentally demonstrated the effectiveness of the new coarse-to-fine strategy, which classifies inner-surface voxels, outer-surface voxels, and surface voxels into different classes. In the introduction section, we claim this strategy works since it solves the problem that the classifier has to waste its capability to bridge the gap between inner-surface voxels and outer-surface voxels. Grouping inner-surface and outer-surface voxels into the same class will force the model to waste its capability only when there is a gap between these two types of voxels. Therefore, the first part of Section 4.3 aims to show this. We provide qualitative and quantitative results to show the difference between inner-surface voxel features and outer-surface voxel features. First, we visualize the feature space of surface, inner-surface and outer-surface voxels using t-SNE. As shown in Figure 3, inner-surface voxels (red points) are far from outer-surface voxels (blue points) in the feature space, thus demonstrating they are quite different. Second, we measure the similarity between inner-surface and outer-surface voxels, the similarity between inner-surface and surface voxels, and the similarity between outer-surface and surface voxels. The result shows that the similarity between inner-surface and outer-surface voxels is relatively lower. This result also demonstrates that inner-surface voxels are different from outer-surface voxels. In addition, we refer the reviewer to Questions 4 and 6 of reviewer xj9n. In response to Question 4 of reviewer xj9n, we conduct experiments to show the accuracy in discriminating voxels. The result shows that only 2.67% and 4.46% of inner and outer voxels are misclassified as outer and inner voxels at the coarse level. This also supports our assumption that there is an intrinsic gap between inner-surface voxels and outer-surface voxels since the model can easily distinguish them in most cases. In response to Question 6 of reviewer xj9n, we conduct another experiment to evaluate the similarity between different types of voxels. The result also supports that inner-surface voxels are different from outer-surface voxels.
>
> **Q3: Could you provide details on the inference time?**
>
> **Re:** IOAR costs 16 minutes and 18 seconds to reconstruct meshes for 100 scenes in the test set of the ScanNet dataset. On average, it costs 9.78 seconds per scene. The time of data loading, data preprocessing, and data postprocessing is included. If we only consider the inference time of the model, IOAR costs 14 minutes and 58 seconds to predict the TSDF volumes for 100 scenes. On average, it costs 8.98 seconds per scene. We also provide details of the inference phase. In the inference phase, the input is a video in which all frames have corresponding camera intrinsic and extrinsic. Based on the camera extrinsic, camera intrinsic, and image resolution, the boundary of the space captured by these images can be estimated. We split the entire space volume into sub-volumes to save memory. Since it is computationally expensive to consider all input frames, we select $N=60$ frames from the video for each sub-volume. We follow the selection method in [9] to reduce redundancy. Specifically, this selection method ensures that the distance between any two frames is far enough and the angle is large enough. These selected frames are input into the CNN backbone to extract image features at different scales. With the help of camera intrinsic and extrinsic, the image features are back-projected to form 3D feature volumes. A transformer module fuses the feature volumes from $N$ frames into a single feature volume. The fused feature volumes of sub-volumes are combined to form the fused feature volume of the global volume. The fused global feature volume is input into the TSDF branch and inner-outer aware occupancy branch to extract the TSDF feature volume and occupancy feature volume. The TSDF and occupancy feature volume at coarse and medium levels will pass to the next level to provide information from the previous scale. At coarse and medium levels, our model predicts the occupancy based on occupancy feature volume and then filters out inner-surface and outer-surface voxels based on the predicted occupancy. Finally, our model predicts the TSDF volume of the scene based on the TSDF feature volume at the fine level and then generates the 3D meshes using the marching cube algorithm[13].
>
>  **Q4: There are some typos in the paper.**
>
> **Re:** We will fix these typos in the camera-ready version.

---

> > ### Comment · Reviewer_Ga1p · 2023-08-21
> > **Thanks for the clarifications**
> >
> > Thank you for addressing the concerns. I agree rewriting some of the write up to learn the misunderstanding (as also pointed by xj9n) would be helpful. I updated my review based on the rebuttal.

---

> > > ### Author Response · Authors · 2023-08-21
> > >
> > > We sincerely thank you for taking the time to review our paper! We are pleased that our response has helped address your concerns.

---

### Official Review · Reviewer_xj9n · 2023-07-03

**Soundness:** 3 good
**Presentation:** 2 fair
**Contribution:** 3 good
**Rating:** 6
**Confidence:** 4

**Summary:**

The paper presents a multi-resolution method for volumetric 3D reconstruction from an input video of a static scene captured by a moving camera. Camera poses must be externally provided. The key observation is that non-surface voxels inside and outside the surfaces have different properties which have been ignored by previous methods for estimating occupancy or the signed distance function. Based on this observation, the paper proposes a three-label classification of space: inner, surface and outer voxels. The claim is that learning decision boundaries between these three types is an easier problem than learning to discriminate between surface and non-surface voxels. This claim is supported by experimental results on widely used data, in comparison with SOTA methods, and by analysis of the classifier.

** Post discussion update**

As I wrote during the discussion, I have been convinced by the other reviews and the authors responses to upgrade my rating.

**Strengths:**

S1. The largest strength of the paper is that the argument that inner and outer voxels with respect to the surface are different makes sense, is novel, and leads to superior quantitative results. My impression is that this distinction is stronger near the surfaces, where it is most useful. I would argue that inner and outer voxels far from the surfaces are just photo-inconsistent, as we used to say, but the critical decisions have to be made near the surface.

S2. Experiments are conducted on large, popular datasets, with carefully analyzed protocols, and the selection of baseline methods is satisfactory. Experimental results demonstrate that inner-outer distinction is powerful, as is the multi-resolution scheme. IOAR, the proposed method, outperforms the baselines by a wide margin on all metrics and all datasets.

S3. Good generalization performance is also shown. This is an important property to enable future deployment of learning-based systems.

**Weaknesses:**

W1. There is a claim that occupancy and TSDF branches interfere with each other. In many other papers, it has been shown that multi-task learning is beneficial when the network learns related tasks. Here, there is a common 3D backbone that generates input for the occupancy and TSDF heads, so it is possible that synergies do exist and the point I am raising is due to lack of clarity. The corresponding ablation study, summarized in Table 4, does not shed enough light on how separate the TSDF and occupancy branches should be. Does the absence of branches mean that occupancy is predicted from the TSDF or vice versa? In any case, the differences are small.

W2. Equation (9), the loss, requires further explanation. Why are occupancy losses applied to the coarse and medium resolution and the TSDF loss to the fine resolution? I assume that this combination worked best in practice, but I would like to see the intuition behind it. Hopefully, the reasons/observations that led to this choice could be useful to other researchers.

W3. Despite criticism of volumetric methods in the abstract, the proposed approach is fully volumetric. Standard techniques to reduce the memory footprint, such as octrees and hashing, are applicable similarly to most other volumetric methods. The criticism in line 4 is unwarranted. (It is also not the best way to introduce the contribution of IOAR.)

W4. Some additional ablation studies would be informative. If the authors already have the relevant data even on parts of the datasets, I would like to see (i) the effects of the multi-resolution scheme compared to a single-resolution implementation, and (ii) the accuracy in discriminating inner vs outer voxels.

W5 (minor). Lines 195-196 contain: “At the fine level, all voxels are treated as potential surface voxels. Therefore, we do not set a fine-level occupancy branch.” This is unclear to me. Conceptually, there should be more empty voxels at a finer resolution since surfaces have zero thickness.

W6 (minor). Lines 286-291 describe an unsatisfactory procedure for measuring the similarity across the three voxel types. Assuming that the distributions are multi-modal, which is likely, representing each category by the average may be a poor approximation. Randomly sampling thousands of representatives from each category would have been better.


Minor Comments

The survey of related work is thorough. I have two suggestions, but their degree of overlap with the proposed method is similar to other methods.

Liu C, Gu J, Kim K, Narasimhan SG, Kautz J. Neural RGB->D sensing: Depth and uncertainty from a video camera. CVPR 2019.

Xie J, Lei C, Li Z, Li LE, Chen Q. Video depth estimation by fusing flow-to-depth proposals. IROS 2020.


34, 117: “marching cubes”

41: “their observations”

42: “there is” and “an intrinsic” – “intrinsical” is not a word and it appears several times in the paper.

58, 198, among other instances: “inter-outer” should be corrected.

86: “a 3D U-Net predicts”

98-105: I disagree that Occupancy Networks (please correct spelling) and DeepSDF focus at novel view synthesis rather than geometric accuracy.

Eqs. (1), (3), (4), (5), (7): large spaces between the two parts of these equations would help.

176: “state” would be better than “situation” here.

179: “operations… are”

193: “the rest procedure” should be corrected.

Many occurrences: there should be a space between a word and a reference in square brackets.

Table 3: the reference to VoRTX is wrong.

Supp. 42: “Broader”

**Questions:**

The distinction between inner and outer voxels is worthy of acceptance, but there are open questions. My preliminary ranking is low, but can be improved if there are satisfactory answers to the first three weaknesses pointed out above.

**Limitations:**

There is a paragraph in the supplement describing three challenging cases for the proposed algorithm. A sentence summarizing difficulties due to occlusion and transparency could have been included in the main paper. There is no potential negative societal impact.

---

> ### Author Rebuttal · Authors · 2023-08-09
>
> **Q1: The difference between using separate or shared TSDF and occupancy branches is not clear.**
>
> **Re:** To clarify the difference, we explain how existing methods and IOAR predict occupancy and TSDF. Conventionally, a shared 3D CNN is used to refine the 3D feature volume. Then the TSDF and occupancy heads (i.e., two MLPs) predict the TSDF and occupancy for each voxel based on the same refined 3D feature volume. We notice that it is difficult for a 3D CNN to learn such versatile features that can provide information for both TSDF and occupancy prediction. Therefore, different from existing methods, IOAR contains two separate 3D CNNs for TSDF and occupancy prediction. We experimentally demonstrate the effectiveness of this design in the ablation study (section 4.2). The full IOAR model with separate 3D CNNs for TSDF and occupancy heads (line 4 of Table 4) outperforms the IOAR variant using a shared 3D CNN for TSDF and occupancy heads (line 3 of Table 4).
>
> **Q2: Why are occupancy losses applied to the coarse and medium resolutions and the TSDF loss to the fine resolution?**
>
> **Re:** The occupancy prediction aims to reduce memory cost by reducing the voxel number in the next stage. Since we filter out non-surface voxels at the end of coarse and medium levels, we only supervise the occupancy prediction in these two resolutions. Similarly, we aim to reconstruct the surface in fine resolution by predicting the TSDF. So, we only supervise the TSDF prediction in fine resolution. We conduct experiments to evaluate different variants. Here, "IOAR" is the original IOAR model; "IOAR all Occ" is a variant with occupancy losses at all resolutions; "IOAR all TSDF" is a variant with TSDF losses at all resolutions; "IOAR all" is a variant with occupancy and TSDF losses at all resolutions. All variants achieve comparable performance except the variant with all TSDF losses drops a bit on Recall.
>
> |Model|Acc|Comp|Prec|Recall|F-score|
> |-|-|-|-|-|-|
> |IOAR|0.043|0.090|0.748|0.597|0.663|
> |IOAR all Occ|0.044|0.093|0.750|0.592|0.660|
> |IOAR all TSDF|0.042|0.100|0.751|0.579|0.652|
> |IOAR all|0.044|0.089|0.743|0.593|0.658|
>
> **Q3: The criticism in line 4 is unwarranted.**
>
> **Re:** Line 4 does not aim to criticize the memory cost of all volumetric-based methods increases cubically. As one of the volumetric-based methods, the memory cost of IOAR also increases cubically. In fact, we aim to explain coarse-to-fine strategy is necessary for volumetric-based methods since their memory cost increase cubically. This is a misunderstanding caused by our writing. We believe rewriting line 4 as "The memory cost of volumetric-based methods will grow cubically as the volume size increases, so a coarse-to-fine strategy is necessary for saving memory." can clear the misunderstanding.
>
> **Q4: I would like to see (i) a single-resolution implementation, and (ii) the accuracy of discriminating voxels.**
>
> **Re:**  For (i), a single-resolution implementation will lead to out-of-memory since the coarse-to-fine framework is necessary for volumetric-based methods [8,9,10,11,12] to reduce memory costs. As an alternative, we design an IOAR variant with only two levels. Without the medium resolution, the performance drops a lot.
>
> |Model|Acc|Comp|Prec|Recall|F-score|
> |-|-|-|-|-|-|
> |two level|0.058|0.096|0.670|0.553|0.604|
> |three level|0.043|0.090|0.748|0.597|0.663|
>
> For (ii), we conduct experiments to evaluate the accuracy of discriminating voxels. At the coarse level, only 2.67% and 4.46% of inner and outer voxels are misclassified as outer and inner voxels. This supports our assumption that outer voxels are quite different from inner voxels since the model can easily distinguish them in most cases. At the medium level, we can observe a lot of inner and outer voxels are predicted as surface voxels. After filtering at the coarse level, most remaining voxels are close to the surface, so their features are similar to surface voxels. As a result, our model misclassifies them into surface voxels.
>
> |Coarse|Inner|Outer|Surface|
> |-|-|-|-|
> |Predict as Inner|71.83%|4.46%|4.17%|
> |Predict as Outer|2.67%|60.17%|0.77%|
> |Predict as Surface|25.50%|35.37%|95.06%|
>
> |Medium|Inner|Outer|Surface|
> |-|-|-|-|
> |Predict as Inner|47.30%|4.73%|6.14%|
> |Predict as Outer|2.48%|48.40%|4.82%|
> |Predict as Surface|50.22%|46.87%|89.04%|
>
> **Q5: Why do not authors set a fine-level occupancy branch to filter out more non-surface voxels?**
>
> **Re:** First, the occupancy prediction at the coarse and medium levels aims to reduce the memory cost at the next level by reducing the voxel number. Since there are no finer levels, we do not set a fine-level occupancy branch. Second, as demonstrated in Q4, these voxels at the fine level are very close to the surface. Therefore, adding another occupancy branch can hardly further filter out non-surface voxels. Finally, the area that does not contain a surface can be filtered out based on the predicted TSDF. Intuitively, if the TSDF value of eight voxels has the same sign, the cubic space with these voxels as its vertices does not contain a surface. We refer the reviewer to the marching cube algorithm [13] for more details.
>
> **Q6: Randomly sampling thousands of representatives from each category is better for measuring the similarity.**
>
> **Re:** We conducted experiments following the advice. In this setting, the similarity between inner and outer voxels is 0.4512, the similarity between inner and surface voxels is 0.5925, and the similarity between outer and surface voxels is 0.5441. These results also support that inner voxels are quite different from outer voxels.
>
> **Q7: There are two works related to your work.**
>
> **Re:** NeuralRGBD and DeepV2D estimate depth for an RGB video, thus related to the depth-based 3D reconstruction methods. We will add the introduction for these works in the related work section.
>
> **Q8: There are some typos in the paper.**
>
> **Re:**  We will fix these typos in the camera-ready version.

---

> > ### Comment · Reviewer_xj9n · 2023-08-14
> > **Thank you for the clarifications**
> >
> > Below are my responses to the authors’ rebuttal. The first three weaknesses were the most important ones to be addressed.
> >
> > I consider the response to W1 satisfactory. I will accept experimental results on the shortcomings of a multi-task implementation over my speculation.
> >
> > The response to W2 is informative. I suggest integrating the first sentences into the paper because the new text seems clearer. I agree with the authors that rewriting the description of the coarse-to-fine approach would also eliminate W3 and W5.
> >
> > The response to W4 is also satisfactory. Tuning the system to favor recall of surface voxels makes sense.
> >
> > The rest of my comments were minor.
> >
> > I have also read the other reviews and responses. I am now more positive towards the paper, but I will wait for the end of the discussion before modifying my recommendation.

---

> > > ### Author Response · Authors · 2023-08-18
> > >
> > > Thank you for your time and efforts in reviewing our paper! Your comments help us improve the quality of this work.

---

### Official Review · Reviewer_seEC · 2023-07-05

**Soundness:** 3 good
**Presentation:** 2 fair
**Contribution:** 3 good
**Rating:** 4
**Confidence:** 3

**Summary:**

This paper modifies previous coarse-to-fine frameworks to classify voxels into outer-surface, inner-surface, and surface voxels. In addition, the TSDF branch is added to further improve the performance. Extensive experiments show the good performance of the proposed method.

**Strengths:**

1. The motivation is interesting.
2. Extensive experiments are conducted and the performance is good.

**Weaknesses:**

1. This paper proposes to predict TSDF and reformulate the occupancy into 3 cases. However, the basic framework is built based on 3D-Former and VoRTX (i.e., coarse-to-fine). Hence, the novelty is somehow limited.
2. In Line 55, it would be confusing to say that "IOAR explores a new coarse-to-fine strategy" since the coarse-to-fine framework is similar to previous works.
3. As a SoTA method, 3D-Former is not compared in the experiments.

Typos:
- Line 140: voRTX -> VoRTX
- Table 2: Vortx [8] -> VoRTX [11]

**Questions:**

See weaknesses.

**Limitations:**

No limitations are mentioned in this paper.

---

> ### Author Rebuttal · Authors · 2023-08-09
>
> **Q1: The novelty of this paper is somehow limited since their model is built based on the coarse-to-fine framework like previous methods (e.g., 3D-Former and VoRTX).**
>
> **Re:**  Using the basic coarse-to-fine framework does not mean the novelty of IOAR is limited. As we have introduced in the abstract and introduction, using the coarse-to-fine framework to reduce the cubically increasing memory cost is necessary for volumetric-based methods [8,9,10,11,12]. After Atlas [8] proposed the basic coarse-to-fine framework, succeeding methods like NeuralRecon[9], TransformFusion [10], VoRTX [11], and 3D-Former [12] are all built based on this basic framework. Although these works are based on the coarse-to-fine framework, their novelty is not limited since they improve the 3D reconstruction in different aspects. So does the proposed IOAR. To make it more specific, we briefly introduce the novelty of IOAR and the mentioned VoRTX and 3D-Former. VoRTX weights the contribution of image features from different views for each voxel to extract more informative feature volumes. 3D-Former design a 3D Transformer to replace the 3D CNN, which provide a stronger 3D backbone for 3D reconstruction. Different from them, IOAR aims to classify the potential surface voxels more accurately with a new coarse-to-fine strategy, which classifies the inner-surface, outer-surface, and surface voxels into different classes. Since VoRTX, 3D-Former, and IOAR improve the 3D reconstruction in different aspects, they are three complementary and orthogonal works with their own novelty.
>
> **Q2: In Line 55, it would be confusing to say that "IOAR explores a new coarse-to-fine strategy" since the coarse-to-fine framework is similar to previous works.**
>
> **Re:** IOAR does explore a new coarse-to-fine strategy. As we have explained in Q1, all volumetric-based methods [8,9,10,11,12] are built based on the basic coarse-to-fine framework, which utilizes a coarse-to-fine strategy to decide which voxels will keep at the next stage. The conventional coarse-to-fine strategy distinguishes potential surface voxels from other voxels and reduces the memory cost by removing the voxels far from the surface. However, this strategy groups the inner-surface voxels and outer-surface voxels into the same class, which neglects the fact that the inner-surface voxels are different from the outer-surface voxels. As a result, the classifier has to spend its capability to bridge this intrinsic gap which can lead to overfitting and thus harm the model generalization. To avoid this problem, IOAR explores a new coarse-to-fine strategy that classifies inner-surface, outer-surface, and surface voxels into different classes. Due to the intrinsic gap, the classifier can easily find the classification boundary between the inner-surface voxels and the outer-surface voxels. Therefore, the classifier can focus on finding the classification boundary between the surface voxels and inner-surface voxels and the classification boundary between the surface voxels and outer-surface voxels. Thanks to the merits of the new coarse-to-fine strategy, IOAR can predict surface voxels more accurately. Our experiments support this claim with both quantitive and qualitative results.
>
> **Q3: As a SoTA method, 3D-Former is not compared in the experiments.**
>
> **Re:** As we have mentioned in the related work section, 3D-Former [12] is an excellent work that designs a 3D Transformer to replace the 3D CNN used by previous works. In their paper, they conducted the ablation study to demonstrate that 3D-Former is a stronger 3D backbone compared with traditional 3D CNN. Therefore, replacing the 3D CNN with the 3D-Former can theoretically improve the performance of any existing volumetric-based 3D reconstruction methods. This is intuitive, just like replacing LeNet with ResNet can theoretically improve the performance of any downstream task. However, all previous methods use 3D CNN as the 3D backbone. Therefore, to have a fair comparison with existing works, we still use the 3D CNN as the 3D backbone. And, since we treat 3D-Former as a stronger backbone, we have not listed it in the experiment tables. An ideal way to make the result more complete is to split the experiment tables into two parts to report the results using the 3D CNN backbone and the results using the 3D-Former backbone. However, reproducing all existing methods using the 3D-Former as the 3D backbone is too computationally expensive. Specifically, training a basic 3D-Former requires 8 V100 GPUs to train 5,000 epochs (costs about 6000 GPU hours), let alone combining it with other methods.
>
> **Q4: There are two typos in the paper.**
>
> **Re:** We will fix these two typos in the camera-ready version.

---

### Official Review · Reviewer_HJrK · 2023-07-06

**Soundness:** 3 good
**Presentation:** 3 good
**Contribution:** 3 good
**Rating:** 6
**Confidence:** 4

**Summary:**

This paper proposes an inner-outer aware reconstruction (IOAR) model for monocular 3D scene reconstruction. Different from existing methods, IOAR can classify the inner-surface voxels and outer-surface voxels, which could lead to better occupancy prediction and TSDF prediction. More specifically it proposes a new inner-outer aware coarse-to-fine strategy and loss, which leads the model to learn to classifier the inner-surface voxels and outer-surface voxels. To avoid the mutual interference between TSDF prediction and occupancy prediction, IOAR separates the occupancy branch from the TSDF branch. Experimental results show IOAR achieves state-of-the-art performance on the large-scale indoor scene dataset ScanNet and can generalize reasonably on unseen scenes.


**Strengths:**

Overall the idea of further classifying exterior and interior surface voxels has merit and seems to generate good results. Separate TSDF and occupancy also make sense. The results are good.

**Weaknesses:**

only scannet is used for testing. would be nice to test on more.

**Questions:**

How about other datasets?

**Limitations:**

the loss function of equation 9 is a sum of all losses, should it be weighted? How should the weights be set to?

---

> ### Author Rebuttal · Authors · 2023-08-09
>
> **Q1: only ScanNet is used for testing. would be nice to test on more.**
>
> **Re:** In the paper, we have tested our model on three different datasets: ScanNet (Tables 1 and 2), ICL-NUIM (Table 3), and TUM-RGBD (Table 3). Following previous methods [11, 12], we evaluate the performance of IOAR on ICL-NUIM and TUM-RGBD datasets with a model trained on the ScanNet dataset. The reason why our model and previous methods test on ICL-NUIM and TUM-RGBD datasets in this manner is ICL-NUIM and TUM-RGBD datasets are too small. Specifically, the ICL-NUIM dataset contains only 8 scenes and the TUM-RGBD dataset contains 13 scenes. On the contrary, the ScanNet dataset contains 1,613 scenes (1,513 for training and 100 for testing). Since the lack of data, training on ICL-NUIM and TUM-RGBD datasets and subsequently testing on the respective datasets may result in poor performance.
>
> We conduct experiments to verify this assumption. We manually split ICL-NUIM and TUM-RGBD datasets, with 6 and 10 scenes for training and 2 and 3 scenes for testing, respectively. The experiment result is worse than expected. With limited data for training, the model failed to learn to reconstruct any surface in the test scenes. That is, no meshes are reconstructed for the test scenes. This result explains why previous methods do not evaluate their model under this setting.
>
> **Q2: the loss function of equation 9 is a sum of all losses, should it be weighted? How should the weights be set to?**
>
> **Re:** Assigning different weights to different losses may result in better performance, while we simply set equal weights to each loss since these losses are in the same order of magnitude. To evaluate the impact of assigning different weights to different losses, we split the losses into three groups: the occupancy losses, the TSDF loss and the projective occupancy losses. We conduct experiments to set different weights to three groups of losses. We denote the weight for occupancy losses as $\lambda_1$, the weight for TSDF loss as $\lambda_2$,  and the weight for projective occupancy losses as $\lambda_3$. So, the weighted loss $\mathcal{L}^W = \lambda_1(\mathcal{L}^c_{OCC} + \mathcal{L}^m_{OCC}) + \lambda_2\mathcal{L}^f_{TSDF} + \lambda_3(\mathcal{L}^c_{P} + \mathcal{L}^m_{P} + \mathcal{L}^f_{P}).$   The results are reported in following table. All variants achieves comparable performance except the one set $\lambda_1$ to 0.5. In summary, assigning different weights to these losses can lead to slight performance variance, especially decreasing the weight of occupancy losses.
>
>
>
> | $\lambda_1$ | $\lambda_2$ | $\lambda_3$ | Acc   | Comp  | Prec  | Recall | F-score |
> | ----------- | ----------- | ----------- | ----- | ----- | ----- | ------ | ------- |
> | 0.5         | 1           | 1           | 0.045 | 0.095 | 0.733 | 0.580  | 0.646   |
> | 1           | 0.5         | 1           | 0.046 | 0.087 | 0.740 | 0.600  | 0.661   |
> | 1           | 1           | 0.5         | 0.044 | 0.090 | 0.744 | 0.597  | 0.661   |
> | 1           | 1           | 1           | 0.043 | 0.090 | 0.748 | 0.597  | 0.663   |

---

### Official Review · Reviewer_5s3L · 2023-07-07

**Soundness:** 4 excellent
**Presentation:** 3 good
**Contribution:** 3 good
**Rating:** 5
**Confidence:** 5

**Summary:**

This paper introduces a novel method for monocular 3D scene reconstruction, termed Inner-Outer Aware Reconstruction (IOAR). In contrast to prior works, IOAR incorporates a unique classification process for outer surfaces, inner surfaces, and surface voxels. Additionally, the method distinctly separates the occupancy branches for IOAR from the TSDF branches, effectively mitigating any mutual interference between them. Notably, the proposed method exhibits exemplary performance across multiple datasets, including ScanNet, ICL-NUIM, and TUM-RGBD. Furthermore, the model, when trained solely on the ScanNet dataset, still achieves superior performance in both ICL-NUIM and TUM-RGBD datasets.

**Strengths:**

- The central concept underlying this paper is both logically sound and remarkably straightforward, yet it proves to be highly effective. The provided ablation study reinforces the efficacy of the proposed module.
- The IOAR method demonstrates unparalleled performance across a range of datasets, indicating its robustness and adaptability.

**Weaknesses:**

- While the contributions are impactful, they are rather direct and simplistic, which could potentially limit the depth and breadth of the paper.
- There is an inconsistency between the numerical values presented in lines 250-251 on page 7 and those in Table 3. For instance, 0.596 is cited in the text, whereas 0.607 is reported in the table. Similar discrepancies are noted for 0.569 (text) vs. 0.564 (table), and 0.488 (text) vs. 0.507 (table). This discrepancy warrants clarification.

**Questions:**

- What is the rationale behind defining the occupancy ground truth to range from 0 to 2? Additionally, how are other values, such as those ranging from -1 to 1, handled in Equation (6)?
- What is the performance of the model when trained on the ICL-NUIM and TUM-RGBD datasets and subsequently tested on the respective datasets?
- It appears that the proposed method continuously learns to discern between the three distinct regions, even during inference. If this assumption is correct, how does the neural network ascertain these three different regions within the voxels from just the initial few frames? It seems plausible that there may not be sufficient occupied 3D voxels to fully comprehend the 3D scene initially. As more frames are encoded into 3D voxels, the neural network may begin to recognize the three regions. Is the information primarily extracted from the visual input or from the coarsely reconstructed 3D voxels?

**Limitations:**

Yes, in supplementary materials.

---

> ### Author Rebuttal · Authors · 2023-08-09
>
> **Q1: While the contributions are impactful, they are rather direct and simplistic, which could potentially limit the depth and breadth of the paper.**
>
> **Re:** The insight behind IOAR is intuitive, but we believe it can inspire future works on 3D reconstruction. The starting point of IOAR is the fact that classifying inner-surface voxels and outer-surface voxels into the same class will force the model to waste its capability to bridge the gap between the two types of voxels. IOAR explores a new coarse-to-fine strategy to solve this problem and achieves performance improvement. The value of IOAR is more than the performance improvement. Existing works improve the performance of 3D reconstruction in various ways, such as adding a transformer mechanism to weigh the features [10, 11] and designing a stronger backbone [12]. On a higher level, they focus on improving 3D reconstruction by increasing the model capability. Different from them, IOAR provides another direction, i.e., to reduce the waste of the model capability. IOAR proves that reducing the waste of model capability is a potential way to improve 3D reconstruction with limited cost. Therefore, IOAR can inspire future works to try to enhance 3D reconstruction by improving the usage of model capability instead of increasing the model capability. We believe this is more valuable than performance improvement since the extensive computational cost is a critical problem in 3D reconstruction, and generally, reducing the waste of model capability requires less computational cost than increasing the model capability.
>
>  **Q2: There is an inconsistency between the numerical values presented in lines 250-251 on page 7 and those in Table 3.**
>
> **Re:** The values presented in Table 3 are correct. We updated the new experiment result in Table 3 while forgetting to update those on page 7. We will fix this problem in the camera-ready version.
>
> **Q3: What is the rationale behind defining the occupancy ground truth to range from 0 to 2? Additionally, how are other values, such as those ranging from -1 to 1, handled in Equation (6)?**
>
> **Re:** Conventionally, the Signed Distance Function (SDF) measures the distance between each voxel and its closest surface. And, the Truncated Signed Distance Function (TSDF) truncated the distance at a given threshold (e.g., 12cm) and normalized the distance to range from -1 to 1 (the sign indicates the inner or outer side). Based on the definition of TSDF, we can easily define the inner-surface, outer-surface, and surface voxels in Equation (6). We define the ground-truth occupancy of inner-surface, surface, and outer-surface voxels as 0, 1, and 2 because assigning ground-truth labels from 0 is conventional in classification tasks. In this case, we can easily indicate the probability that our model makes the correct prediction using a subscript (e.g., in Equation (8)). Theoretically, the ground-truth occupancy of inner-surface, surface, and outer-surface voxels can be defined as any discrete numbers (e.g., -1, 0, 1).
>
> **Q4: What is the performance of the model when trained on the ICL-NUIM and TUM-RGBD datasets and subsequently tested on the respective datasets?**
>
> **Re:** Following previous methods [11,12], we evaluate ICL-NUIM and TUM-RGBD datasets with the model trained on the ScanNet dataset. The reason why previous methods and our model tested on ICL-NUIM and TUM-RGBD datasets using a model trained on ScanNet dataset is ICL-NUIM and TUM-RGBD datasets are too small. Specifically, the ICL-NUIM dataset contains only 8 scenes and the TUM-RGBD dataset contains only 13 scenes. On the contrary, the ScanNet dataset contains 1,613 scenes (1,513 for training and 100 for testing). Because of the lack of data, training on ICL-NUIM and TUM-RGBD datasets and subsequently testing on the respective datasets may result in poor performance.
>
> We conduct experiments to verify this assumption. We manually split ICL-NUIM and TUM-RGBD datasets, with 6 and 10 scenes for training and 2 and 3 scenes for testing, respectively. The experiment result is worse than expected. With limited data for training, the model failed to learn to reconstruct any surface in the test scenes. That is, no meshes are reconstructed for the test scenes. This result explains why previous methods do not evaluate their model under this setting.
>
> **Q5: How does the neural network ascertain these three different regions within the voxels from just the initial few frames in the inference phase? Is the information primarily extracted from the visual input or from the coarsely reconstructed 3D voxels?**
>
> **Re:** In the inference phase, our model does not have to ascertain three different regions from just the initial few frames. Following previous works [8,9,10,11,12], given an input video, we first select $N$ ($N=60$ in the inference time) frames from the entire video. We follow the selection method in [9] to reduce redundancy, which ensures the distance and the angle between any two selected frames are large enough. Image features of all the selected frames will be utilized to generate the 3D feature volumes. Therefore, Our model can consider the information from all selected frames to predict occupancy (i.e., whether a voxel belongs to the inner region, outer region, or on the surface). Since images are the origin of all information, we believe the information is primarily extracted from the visual input.

---

> > ### Comment · Reviewer_5s3L · 2023-08-18
> > **Thank you for the reuttal**
> >
> > After carefully reading the rebuttal, I appreciate the clarifications provided. I have decided to keep my initial rating.

---

> > > ### Author Response · Authors · 2023-08-18
> > >
> > > We appreciate you taking the time and effort to review our paper! Your comments help us improve the clarity and presentation of our paper.

---

### Comment · Area_Chair_VQeG · 2023-08-20
**Thanks for the rebuttal!**

Dear authors,

A quick note to express our gratitude for your timely rebuttal. Your clarifications and responses greatly aid the review process.

Thank you,
Area Chairs

---

### Decision · Program_Chairs · 2023-09-21

**Decision:**

Accept (poster)

**Comment:**

In this paper, the authors propose a new method for monocular 3D scene reconstruction. Multiple reviewers find the insight on exterior vs. interior surface voxels interesting and novel, while some questions requesting further explanation and empirical evidence were raised. The majority of these were addressed by rebuttal. After a careful evaluation of the reviews and rebuttal, the AC agrees with the assessment and recommends acceptance.